

# Enteroaggregative *Escherichia coli* is associated with antibiotic resistance and urinary tract infection symptomatology

Verónica I. Martínez-Santos[1], María Ruíz-Rosas[2], Arturo Ramirez- Peralta[3], Oscar Zaragoza García[4], Luis Armando Resendiz-Reyes[4], Obed Josimar Romero-Pineda[4] and Natividad Castro-Alarcón[4]

[1] Cátedras CONACyT, Universidad Autónoma de Guerrero, Chilpancingo, Guerrero, México
[2] Laboratorio Clínico, Área Microbiología, Clínica Hospital ISSSTE Chilpancingo, Chilpancingo, Guerrero, México
[3] Laboratorio de Patometabolismo Microbiano, Facultad de Ciencias Químico Biológicas, Universidad Autónoma de Guerrero, Chilpancingo, Guerrero, México
[4] Laboratorio de Investigación en Microbiología, Facultad de Ciencias Químico Biológicas, Universidad Autónoma de Guerrero, Chilpancingo, Guerrero, México

Corresponding author
Natividad Castro-Alarcón,
natividadcastro24@gmail.com

## ABSTRACT

**Background**. Uropathogenic *Escherichia coli* (UPEC) is the causative agent of uncomplicated urinary tract infections (UTIs) in ambulatory patients. However, enteroaggregative *E. coli* (EAEC), an emergent bacterial pathogen that causes persistent diarrhoea, has recently been associated with UTIs. The aim of this study was to determine the frequency of EAEC virulence genes, antibiotic resistance, as well as biofilm production of UPEC isolates obtained from ambulatory patients with non-complicated UTIs that attended to the ISSSTE clinic in Chilpancingo, Guerrero, Mexico, and correlate these with the patients' urinary tract infection symptomatology.
**Methods**. One hundred clinical isolates were obtained. The identification of clinical isolates, antimicrobial susceptibility testing, and extended spectrum beta-lactamases (ESBLs) production were performed using the Vitek automated system. Assignment of *E. coli* phylogenetic groups was performed using the quadruplex phylo-group assignment PCR assay. UPEC virulence genes (*hlyA*, *fimH*, *papC*, *iutA*, and *cnf1*) and EAEC virulence genes (*aap*, *aggR*, and *aatA*) were detected by multiple PCR.
**Results**. We found that 22% of the isolates carried the *aggR* gene and were classified as UPEC/EAEC. The main phylogenetic group was B2 (44.1% were UPEC and 77.27% UPEC/EAEC isolates, respectively). Over half of the UPEC/EAEC isolates (63.64%) were obtained from symptomatic patients, however the *aatA* gene was the only one found to be associated with the risk of developing pyelonephritis (OR = 5.15, $p = 0.038$). A total of 77.71% of the UPEC/EAEC isolates were ESBL producers and 90.91% multidrug-resistant (MDR). In conclusion, UPEC/EAEC isolates are more frequent in symptomatic patients and the *aatA* gene was associated with a higher risk of developing pyelonephritis, along with UPEC genes *hlyA* and *cfn1*. UPEC/EAEC isolates obtained from UTI showed ESBL production and MDR.

## INTRODUCTION

Urinary tract infections (UTIs), defined as the presence of $\geq 10^5$ colony-forming units (CFU) per mL of midstream urine, are one of the most common bacterial infections, affecting around 150 million people worldwide per year (*Flores-Mireles et al., 2015*; *Smelov, Naber & Johansen, 2016*). In Mexico, these infections were the third cause of morbidity in 2017, with 4,474,599 cases, and an incidence of 3,622.62/100,000 inhabitants. The most affected age group was 25 to 44 years old, with 1,344,198 cases, of which 1,094,069 were in women and 250,129 in men (*Salud, 2017*). UTIs occur mainly in women, of which around 60% are estimated to have an UTI at least once in their lifetime (*Smelov, Naber & Johansen, 2016*). Although women are affected in a ratio 8:1 with respect to men, these infections are also an important cause of morbidity in young boys and older men (*Flores-Mireles et al., 2015*). Some medical conditions can predispose or favor the occurrence of UTIs, like type 2 diabetes, which seems to increase the incidence of symptomatic infections (*Nicolle, 2005*; *Hamdan et al., 2015*); and pregnancy, which makes women more sensitive to this infection (*Delzell Jr & Lefevre, 2000*). UTIs are classified as uncomplicated or complicated depending on the absence or presence, respectively, of structural abnormalities on the urinary tract (*Zacche & Giarenis, 2016*). These infections can present as asymptomatic bacteriuria, which is defined by the presence of a positive urine culture with $10^5$ CFU/mL but without symptoms, or as symptomatic infections, which produce dysuria with or without frequency, urgency, and pain (*Geerlings, 2016*).

UTIs are primarily caused by *Escherichia coli*, a Gram-negative bacterium that is part of the intestinal microbiota. This bacterium is the main etiological agent of UTIs, being responsible for 75–95% of the cases (*Gupta et al., 2011*). Although *E. coli* is usually a commensal organism, there are several pathotypes that cause intestinal and extra intestinal diseases. *E. coli* strains that cause intestinal diseases are known as diarrhoeagenic *E. coli* (DEC), and are classified into 6 different pathotypes, named: enteropathogenic *E. coli* (EPEC), enterohaemorrhagic *E. coli* (EHEC), enterotoxigenic *E. coli* (ETEC), enteroinvasive *E. coli* (EIEC), diffusely adherent *E. coli* (DAEC), and enteroaggregative *E. coli* (EAEC). Extra-intestinal *E. coli* (ExPEC), on the other hand, include avian pathogenic *E. coli* (APEC), meningitis/sepsis-associated *E. coli* (MNEC), and uropathogenic *E. coli* (UPEC) (*Kaper, Nataro & Mobley, 2004*).

UPEC is the main etiological agent of UTIs, both community-acquired and nosocomial, causing around 90% and 50% of the cases, respectively; followed by *Klebsiella pneumoniae*, *Enterococcus faecalis*, *Proteus mirabilis*, *Pseudomonas aeruginosa*, and *Staphylococcus aureus*, among others (*Flores-Mireles et al., 2015*). The virulence potential of UPEC is determined largely by the presence of several virulence factors, among which are adhesins, toxins, and siderophores. Some of the virulence factors that help UPEC to overcome host defenses and establish infection are type I and P fimbriae (encoded by the *fim* and *pap* operons, respectively), the alpha-haemolysin (encoded by *hlyA*), the ferric aerobactin receptor (encoded by *iutA*), and the cytotoxic necrotizing factor 1 (encoded by *cnf1*) (*Johnson, 1991*).

In recent years, the diarrheagenic pathotype EAEC has also been identified as a causative agent of UTIs (*Park et al., 2009*; *Olesen et al., 2012*; *Herzog et al., 2013*; *Toval et al., 2013*; *Salmani et al., 2016*). EAEC is usually the etiological agent of acute and persistent diarrhea in developing countries, also affecting people who travel to these countries, causing traveler's diarrhea (*Harrington et al., 2005*). The main molecular characteristic of EAEC is the presence of the virulence plasmid pAA. This plasmid carries predictor genes *aatA* (previously known as CVD432 probe, which codes for a TolC-like outer membrane protein) and *aggR* (which codes for the master regulator AggR); along with other EAEC genes, like *aap* (dispersin gene), *pet* (plasmid-encoded toxin gene) and several aggregative adherence fimbriae genes. However, given the genetic plasticity of EAEC, some strains carry plasmids with different number and combination of genes (*Nishi et al., 2003*; *Lara et al., 2017*).

Phylogenetically, *E. coli* can be divided into eight phylogroups, of which seven are *E. coli sensu stricto* (A, B1, B2, C, D, E, F), and one is the *Escherichia* cryptic clade I (*Clermont, Christenson JK & Gordon, 2013*). Extraintestinal *E. coli* strains, including UPEC, belong mainly to phylogroups B2 and D, whereas EAEC strains have been found to be in phylogroups A, B1, B2, and D, suggesting that EAEC strains are phylogenetically diverse and originate from multiple lineages (*Escobar-Paramo et al., 2004*; *Imuta et al., 2016*). The aim of this study was to determine the frequency of EAEC virulence genes, antibiotic resistance, as well as biofilm production of UPEC isolates obtained from ambulatory patients with non-complicated UTIs that attended to the ISSSTE clinic in Chilpancingo, Guerrero, Mexico, and correlate these with the patients' urinary tract infection symptomatology.

## MATERIALS AND METHODS

### *E. coli* isolates, antibiotic susceptibility testing, and ESBL production

Samples were collected from November 2016 to March 2017 and only one isolate per patient was examined. Urine samples from 100 ambulatory patients with community acquired UTIs attending the ISSSTE clinic in Chilpancingo, Guerrero, Mexico where analyzed. The study was approved by the Research Ethics Committee of the Autonomous University of Guerrero (CB-002/2021) and the Ethics Committee of the ISSSTE clinic. All participants agreed to participate and gave their informed consent in writing.

The urine cultures were processed using conventional methods and included samples with a viable count of $>10^5$ CFU/mL. The antibiotics assayed in the susceptibility test were ampicillin, ampicillin/sulbactam, cefazolin, ceftriaxone, cefepime, aztreonam, amikacin, gentamicin, tobramycin, ciprofloxacin, nitrofurantoin, and trimethoprim/sulfamethoxazole. Isolates with a resistance to three or more antibiotics were classified as multidrug-resistant (MDR). The identification of clinical isolates, antimicrobial susceptibility testing, and ESBLs production detection were performed using the Vitek2 automated system (BioMérieux) and AST-GN70 cards (Lot Number 590357210), and biochemical test.

### Conventional phylogenetic grouping

Total DNA extraction and assignment of *E. coli* phylogenetic groups (A, B1, B2, C, D, E, F and Clade I) by the quadruplex phylo-group assignment PCR assay described

**Table 1** Oligonucleotides used in this study.

| Gene | Oligonucleotide sequence (5′–3′) | Product size | Reference |
|---|---|---|---|
| *hlyA* | AACAAGGATAAGCACTGTTCTGGCT<br>ACCATATAAGCGGTCATTCCCGTCA | 1,177 | *Nishi et al. (2003)* |
| *fimH* | TGCAGAACGGATAAGCCGTGG<br>GCAGTCACCTGCCCTCCGGTA | 508 | *Olesen et al. (2012)* |
| *papC* | GTGGCAGTATGAGTAATGACCGTTA<br>ATATCCTTTCTGCAGGGATGCAATA | 200 | *Olesen et al. (2012),<br>Park et al. (2009)* |
| *iutA* | GGCTGGACATCATGGGAACTGG<br>CGTCGGGAACGGGTAGAATCG | 300 | *Salmani et al. (2016)* |
| *cnf1* | AAGATGGAGTTTCCTATGCAGGAG<br>CATTCAGAGTCCTGCCCTCATTATT | 498 | *Nishi et al. (2003)* |
| *aap* | CTTGGGTATCAGCCTGAATG<br>AACCCATTCGGTTAGAGCAC | 310 | *Salud (2017)* |
| *aggR* | CTAATTGTACAATCGATGTA<br>AGAGTCCATCTCTTTGATAAG | 457 | *Salud (2017)* |
| *aatA* | CTGGCGAAAGACTGTATCAT<br>CAATGTATAGAAATCCGCTGTT | 629 | *Salud (2017)* |

by *Clermont, Christenson JK & Gordon (2013)* were performed as described previously (*Hernandez-Vergara et al., 2016*).

## Detection of virulence genes of the isolates

UPEC virulence genes *hlyA* (alpha-hemolysin), *fimH* (type 1 fimbriae), *papC* (P-fimbriae), *iutA* (ferric aerobactin receptor), and *cnf1* (cytotoxic necrotizing factor 1) were detected by PCR using primers listed in Table 1. The reaction was performed in a final volume of 25 µL, containing 12.5 µL Go Taq Green Master Mix (Promega), 100 ng of DNA, and 0.6 µM of each primer. The amplification conditions were: 1 cycle at 94 °C for 5 min, 25 cycles at 94 °C for 30 s, 63 °C for 30 s, and 72 °C for 3 min, and 1 cycle at 72 °C for 10 min in a thermal cycler (Bio-Rad). The amplified products were analyzed by agarose gel electrophoresis (1.5%) stained with ethidium bromide and visualized in an ultraviolet transilluminator.

EAEC virulence genes *aap* (dispersin), *aggR* (transcriptional regulator), and the AA probe were detected by PCR using primers listed in Table 1. The reaction was performed in a final volume of 12.5 µL containing 100 ng DNA, 15 pmol of each oligonucleotide, and 5 µL of Go Taq Green Master Mix (Promega). The conditions used were 1 cycle at 94 °C for 5 min, 25 cycles at 94 °C for 30 s, 63 °C for 30 s and 72 °C for 3 min, and a final cycle at 72 °C for 10 min in a thermal cycler (Bio-Rad). The amplified products were analyzed by polyacrylamide gel electrophoresis (6%) stained with silver nitrate.

## Biofilm quantification

Biofilm formation was determined by the quantitative method in 96-well polystyrene microplates. Briefly, 3 colonies were cultured in three mL of BHI broth and incubated for 2 h at 37 °C. The optical density (OD) of the culture was measured at 630 nm, fresh BHI
media was inoculated adjusting the inoculum to an $OD_{630}$ of 0.15, and incubated for 24 h at 37 °C. After incubation, 20 μL of each culture were added to each well containing 180 μL of LB broth supplemented with 5% glucose. The microplate was then incubated for 24 h at 30 °C, and the bacterial growth was quantified at 630 nm using a microplate reader. For biofilm quantification, the supernatants were removed, and the wells were washed twice with sterile PBS 5 mM and dried at room temperature for 30 min and 200 μL of crystal violet 1% was added for 15 min. The dye was removed, and the wells were washed again twice with PBS 5 mM and dried for 20 min. Then, 200 μL of 96% ethanol were added for 10 min. Quantification was performed at a wavelength of 570 nm in a microplate reader (Multiskan Go, Thermo Scientific). The results are the average of three independent experiments. The isolates were classified as low and high biofilm producers (*Stepanovic et al., 2007*).

## Clinical characteristics

The classification of patients according to the type infections was performed considering the clinical symptomatology. Cystitis was identified by typical clinical symptoms, including dysuria, frequent voiding, and lower abdominal pain, whereas pyelonephritis was clinically identified by fever, nausea, dysuria, urgent voiding, flank pain, and lumbar tenderness (*Firoozeh et al., 2014*).

## Statistical analysis

The statistical analyses were performed using the STATA V.13.0 software and GraphPad Prims V.8.0. The frequencies were determined by Chi-square test and Fisher's exact test. The regression model adjusted by age and gender were determined in the UPEC and EAEC with the virulence genes and clinic symptomatology. Odds ratios (ORs), 95% confidence intervals (CI), and $p$ values were calculated. The $p < 0.05$ were considered statistically significant.

# RESULTS

## Phylogenetic groups and presence of EAEC isolates among UPEC

A total of 100 clinical isolates were tested. PCRs were performed in order to determine the phylogenetic group and to detect the virulence genes of each isolate. One of the isolates was identified as *E. albertii*, so it was discarded. Of the 99 remaining isolates, 22 were classified as UEPC/EAEC due to the presence of the *aggR* gene, while the rest were classified as UPEC. When the isolates were grouped according to the phylogenetic group, we found that most of the isolates, both UPEC and UPEC/EAEC, belonged to the B2 group (44.1% and 77.27%, respectively) (Fig. 1).

## Occurrence of UPEC and UPEC/EAEC in patients with different clinical symptoms and demographic characteristics

The distribution of the isolates was then analyzed according to the clinical and demographic characteristics of the population in order to determine if there was a relation between these characteristics and the pathotypes (Table 2). As expected, most of the isolates (76.77%) were obtained from female patients, both UPEC and UPEC/EAEC. Regarding patients' age,

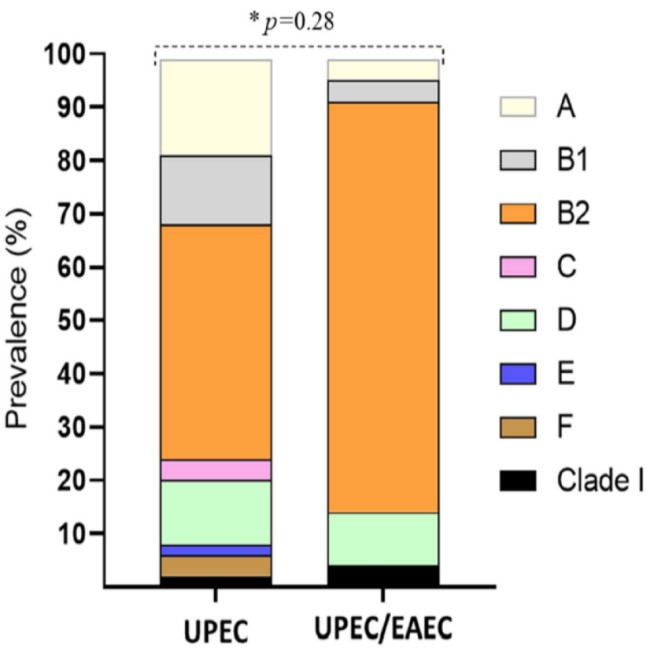

**Figure 1   Distribution of phylogenetic groups of UPEC and UPEC/EAEC isolates.** Data in percentage. An asterisk (*) indicates Fisher's exact test. $p$ value <0.05 was considered statistically significant.

59% of the UPEC/EAEC isolates were obtained from older adults, while 48% of the UPEC isolates were obtained from elderly patients. Of all the isolates, 66.67% were obtained from patients with no UTI-associated risk factors. The most frequent risk factor in our population was type 2 diabetes (T2D), followed by T2D/chronic renal failure, chronic renal failure, catheter use and pregnancy. No significant difference was found in the number of UPEC/EAEC and UPEC isolates obtained. Interestingly, most of the UPEC/EAEC isolates (63.64%) were isolated from patients with clinical symptomatology, while most of the UPEC isolates (59.74%) were obtained from asymptomatic patients. This result suggests that UPEC/EAEC virulence genes seem to be associated with the presence of clinical symptoms. However, when we grouped the isolates according to the type of infection, the same percentage of UPEC/EAEC isolates were isolated from patients with asymptomatic bacteriuria and cystitis (36.36%).

## Occurrence of virulence genes in EAEC and UPEC isolated from asymptomatic and symptomatic patients

Since apparently UPEC/EAEC isolates are more prevalent in patients with clinical symptoms, we wondered if any of the virulence genes detected could contribute to the clinical symptomatology. As seen in Table 3, the *aggR* gene was found to be more predominant in isolates obtained from patients with clinic symptomatology (31%), while the gen *aapA* was the less frequent (24%); however, only the gen *aatA* is more frequent in isolates from patients with clinical symptoms than without symptoms ($p = 0.045$). Regarding UPEC virulence genes, *hlyA* and *cfn1* were also found to be more frequent in

**Table 2 Clinical and demographic characteristics of the population associated with *E. coli* pathotypes.**

| Variables | Total $n = 99$ (%) | UPEC $n = 77$ (%) | UPEC/EAEC $n = 22$ (%) | p value [*] |
|---|---|---|---|---|
| **Sex** | | | | |
| Male | 23 (23.33) | 17 (22.08) | 6 (27.27) | 0.58 |
| Female | 76 (76.77) | 60 (77.92) | 16 (72.73) | |
| **Age (years)** | | | | |
| Underage (0-17) | 4 (4.04) | 4 (5.19) | 0 | 0.22 |
| Young adults (18–34) | 7 (7.07) | 7 (9.09) | 0 | |
| Older adults (35–59) | 42 (42.42) | 29 (37.66) | 13 (59.09) | |
| Elderly ($\geq$60) | 46 (46.46) | 37 (48.05) | 9 (40.91) | |
| **UTIs-associated risk factors** | | | | |
| None | 66 (66.67) | 53 (68.83) | 13 (59.09) | 0.63 |
| Type 2 Diabetes | 21 (21.21) | 14 (18.18) | 7 (31.82) | |
| Type 2 Diabetes-Chronic renal failure | 5 (5.05) | 4 (5.19) | 1 (4.55) | |
| Chronic renal failure | 3 (3.03) | 2 (2.60) | 1 (1.55) | |
| Catheter | 3 (3.03) | 3 (3.90) | 0 | |
| Pregnancy | 1 (1.01) | 1 (1.30) | 0 | |
| **Clinic symptomatology** | | | | |
| Asymptomatic | 54 (54.55) | 46 (59.74) | 8 (36.36) | 0.08 |
| Symptomatic | 45 (45.45) | 31 (40.26) | 14 (63.64) | |
| **Infections** | | | | |
| Asymptomatic bacteriuria | 54 (54.55) | 46 (59.74) | 8 (36.36) | 0.15 |
| Cystitis | 25 (25.25) | 17 (22.08) | 8 (36.36) | |
| Pyelonephritis | 20 (20) | 14 (18.18) | 6 (27.27) | |

Notes.
Data in n (percentage).
*Fisher's exact test.
$p < 0.05$ was considered statistically significant.

isolates from symptomatic patients when compared to the asymptomatic ones ($p = 0.005$ and 0.020).

Then we wanted to determine if these genes were associated to clinical symptoms of cystitis or pyelonephritis. As seen in Table 4, none of the EAEC virulence genes were associated with symptoms of either infection, however the *aatA* gene seems to increase the risk of developing pyelonephritis (OR = 5.15, 95% CI [1.09–24.32], $p = 0.038$), while UPEC virulence genes *cnf1* and *hlyA* are associated with clinical symptoms of pyelonephritis (OR = 4.22, 95% CI [1.30–13.64], $p = 0.016$ and OR = 8.25, 95% CI [1.83–37.21], $p = 0.006$, respectively).

We also wanted to determine if the production of biofilm was associated with these symptoms, but only 21% (21/99) of the isolates were high biofilm producers, and of these, most of the isolates (47.6%, 10/21) were isolated from patients with asymptomatic bacteriuria.
**Table 3** Distribution of EAEC and UPEC virulence genes in strains isolated from asymptomatic and symptomatic patients.

|  | Total $n = 99$ (%) | Asymptomatic $n = 54$ (%) | Symptomatic $n = 45$ (%) | *p* value[*] |
|---|---|---|---|---|
| **EAEC genes** |  |  |  |  |
| *aatA* | 14 (14) | 4 (7) | 10 (22) | 0.045 |
| *aggR* | 22 (22) | 8 (15) | 14 (31) | 0.088 |
| *aapA* | 23 (23) | 12 (22) | 11 (24) | 0.815 |
| **UPEC genes** |  |  |  |  |
| *papC* | 37 (37) | 17 (31) | 20 (44) | 0.214 |
| *iutA* | 66 (67) | 36 (67) | 30 (67) | 1.000 |
| *fimH* | 84 (85) | 44 (81) | 40 (89) | 0.402 |
| *cnf1* | 24 (24) | 8 (15) | 16 (36) | 0.020 |
| *hlyA* | 15 (15) | 3 (6) | 12 (27) | 0.005 |

Notes.
Data in n (percentage).
[*]Fisher's exact test.
$p < 0.05$ was considered statistically significant.

**Table 4** Association between EAEC and UPEC virulence genes and clinical infection.

| EAEC genes | Total $n = 99$ (%) | AB $n = 54$ (%) | Cystitis $n = 25$ (%) | Pyelonephritis $n = 20$ (%) | *p* value[*] | Cystitis OR, CI 95%, *p* value[a] | Pyelonephritis OR, CI 95%, *p* value[a] |
|---|---|---|---|---|---|---|---|
| *aatA+* | 14 (14) | 4 (7) | 5 (20) | 5 (25) | 0.077 | 4.43 (0.92–21.29) 0.063 | 5.15 (1.09–24.32) 0.038 |
| *aggR+* | 22 (22) | 8 (15) | 8 (32) | 6 (30) | 0.151 | 2.99 (0.85–10.39) 0.085 | 2.95 (0.82–10.50) 0.095 |
| *aapA+* | 23 (23) | 12 (22) | 8 (32) | 3 (15) | 0.428 | 1.51 (0.48–4.74) 0.473 | 0.62 (0.15–2.55) 0.516 |
| **UPEC genes** |  |  |  |  |  |  |  |
| *papC+* | 37 (37) | 17 (31) | 9 (36) | 10 (55) | 0.175 | 0.81 (0.27–2.41) 0.707 | 2.30 (0.79–6.73) 0.126 |
| *iutA+* | 66 (67) | 36 (67) | 16 (64) | 14 (70) | 0.920 | 0.87 (0.30–2.50) 0.806 | 1.36 (0.43–4.23) 0.591 |
| *fimH+* | 84 (85) | 44 (81) | 22 (88) | 18 (90) | 0.603 | 1.42 (0.33–6.14) 0.633 | 1.70 (0.32–899) 0.532 |
| *cnf1+* | 24 (24) | 8 (15) | 7 (28) | 9 (45) | 0.026 | 2.08 (0.63–6.82) 0.224 | 4.22 (1.30–13.64) 0.016 |
| *hlyA+* | 15 (15) | 3 (6) | 5 (20) | 7 (35) | 0.004 | 3.71 (0.78–17.48) 0.097 | 8.25 (1.83–37.21) 0.006 |
| **Biofilm production** |  |  |  |  |  |  |  |
| High | 21 (21) | 10 (19) | 5 (21) | 6 (32) | 0.564 | 1.29 (0.36–4.56) 0.701 | 2.06 (0.61–6.92) 0.243 |

Notes.
Data in n (percentage).
[*]Fisher's exact test.
$p < 0.05$ was considered statistically significant.
[a]Multivariate analysis adjusted by age and gender. Reference category was AB infection presence and negative gene presence.
AB, asymptomatic bacteriuria.

## Resistance pattern, ESBL and biofilm production

The antibiotic susceptibility pattern of the obtained isolates was also analyzed. We found that 46.46% of our isolates were ESBL producers, and this result correlated with the resistance to cefazolin (52.53%), ceftriaxone (47.47%), cefepime (45.45%), and aztreonam (47.47%) (Table 5). The number of UPEC/EAEC isolates resistant to these antibiotics was higher than that of UPEC isolates. Also, there was statistically significant difference between UPEC/EAEC and UPEC in the resistance to tobramycin ($p = 0.027$) and ciprofloxacin ($p = 0.040$). However, there was no statistically significant difference in the resistance

**Table 5  Antibiotic resistance, ESBL production and biofilm production of clinical isolates.**

| Antibiotics | | Total $n = 99$ (%) | UPEC $n = 77$ (%) | UPEC/EAEC $n = 22$ (%) | p value[*] |
|---|---|---|---|---|---|
| Ampicillin | R | 80 (80.81) | 61 (79.22) | 19 (86.36) | 0.55 |
| | S | 19 (19.19) | 16 (20.78) | 2 (13.64) | |
| Ampicillin/Sulbactam | R | 70 (70.71) | 52 (67.53) | 18 (81.81) | 0.28 |
| | S | 29 (29.29) | 25 (32.47) | 4 (18.18) | |
| Cefazoline | R | 52 (52.53) | 35 (45.45) | 17 (77.27) | 0.014 |
| | S | 47 (47.47) | 42 (54.55) | 5 (22.73) | |
| Ceftriaxone | R | 47 (47.47) | 30 (38.96) | 17 (77.27) | 0.002 |
| | S | 52 (52.53) | 47 (61.04) | 5 (22.73) | |
| Cefepime | R | 45 (45.45) | 28 (36.36) | 17 (77.27) | 0.001 |
| | S | 54 (54.55) | 49 (63.64) | 5 (22.73) | |
| Aztreonam | R | 47 (47.47) | 30 (38.96) | 17 (77.27) | 0.002 |
| | S | 52 (52.53) | 47 (61.04) | 5 (22.73) | |
| Amikacin | R | 2 (2.02) | 1 (1.30) | 1 (4.55) | 0.39 |
| | S | 97 (97.98) | 76 (98.70) | 21 (95.45) | |
| Gentamicin | R | 38 (38.38) | 27 (35.06) | 11 (50) | 0.22 |
| | S | 61 (61.62) | 50 (64.94) | 11 (50) | |
| Tobramycin | R | 45 (45.45) | 30 (38.96) | 15 (68.18) | 0.027 |
| | S | 54 (54.55) | 47 (61.04) | 7 (31.82) | |
| Ciprofloxacin | R | 67 (67.68) | 48 (62.34) | 19 (86.36) | 0.040 |
| | S | 32 (32.32) | 29 (37.66) | 3 (13.64) | |
| Nitrofurantoin | R | 17 (17.17) | 14 (18.18) | 3 (13.64) | 0.75 |
| | S | 82 (82.83) | 63 (81.82) | 19 (86.36) | |
| Trimethoprim/Sulfamethoxazole | R | 60 (60.61) | 47 (61.04) | 13 (59.09) | 1.00 |
| | S | 39 (39.39) | 30 (38.96) | 9 (40.91) | |
| ESBL | Negative | 53 (53.54) | 48 (62.34) | 5 (22.73) | 0.001 |
| | Positive | 46 (46.46) | 29 (37.66) | 17 (77.27) | |
| MDR | ≤2 drugs | 30 (30.30) | 28 (36.36) | 2 (9.09) | 0.017 |
| | ≥3 drugs | 69 (69.70) | 49 (63.64) | 20 (90.91) | |
| Biofilm | Low | 74 (77.89) | 58 (77.33) | 16 (80) | 0.532 |
| | High | 21 (22.11) | 17 (22.67) | 4 (20) | |

**Notes.**
Data in n (percent).
[*]Fisher's exact test.
$p < 0.05$ was considered statistically significant.

to aminoglycosides (amikacin and gentamycin) and trimethoprim/sulfamethoxazole. Seventeen percent of the isolates causing UTI were resistant to nitrofurantoin. Interestingly, more UPEC/EAEC isolates were shown to be MDR than UPEC isolates ($p = 0.017$). Regarding biofilm production, only 22.11% of the isolates were high producers.

## DISCUSSION

In this work, we demonstrated the importance of UPEC/EAEC isolates as the causative agent of community acquired UTI. Our results show that of 99 isolates obtained, 22 had the

*aggR* EAEC gene, and so were considered UPEC/EAEC isolates. Previous reports have found a prevalence of hybrid isolates of 3.5% based on the detection of the *aatA* and *pap* genes (*Abe et al., 2008*; *Lara et al., 2017*). The phylogenetic analysis revealed that the UPEC/EAEC isolates were distributed in phylogroups A, B1, B2 and D, which is the distribution reported for EAEC strains (*Imuta et al., 2016*). Most of the UPEC/EAEC isolates obtained in this work belonged to group B2. Our results show that the hybrid isolates obtained in this work are more diverse than those obtained from an outbreak in Copenhagen, Denmark in 1991, which belonged only to phylogroup A (*Olesen et al., 2012*), and those recovered from Brazilian patients with UTI and bacteremia, which belonged only to phylogroups D and A (*Lara et al., 2017*). On the other hand, the UPEC isolates were found to be distributed in all the phylogenetic groups. Although it is considered that ExPEC strains belong to groups B2 and D, and commensal strains are gathered in groups A and B1, we found relatively high percentages of UPEC isolates in groups A and B1, and lower percentages in groups C, E, F, and Clade I, showing that our isolates are highly diverse.

In regard to the clinical and demographic characteristics of our population, none of them seemed to favor the infection by UPEC/EAEC isolates over UPEC isolates, since the frequency of both was similar in both sexes, as well as in patients of different age groups and with different risk factors. Nonetheless, we observed that the frequency of UPEC/EAEC isolates was slightly higher in patients with clinical symptomatology than the UPEC isolates, which were more frequent in asymptomatic patients. The presence of hybrid UPEC/EAEC strains in outbreaks of UTI has led to the assumption that EAEC virulence genes contribute to the virulence of the urinary strains (*Lara et al., 2017*). According to this, most of the UPEC/EAEC isolates were obtained from symptomatic patients, however, we could not relate UPEC/EAEC isolates with a specific infection.

With respect to the contribution of EAEC genes *aap*, *aggR*, and *aatA* to the virulence of the bacteria, it has been shown that these genes are present in EAEC isolates obtained from patients with diarrhea in frequencies similar or higher to the ones we found (*aap* 27–32%, *aatA* 16–33%, *aggR* 11–67%), but they are absent or in low frequencies in EAEC isolates obtained from patients without diarrhea, supporting their role on the developing of the disease. Accordingly, the presence of two or three of these genes has been associated with diarrhea (*Huang et al., 2007*; *Mohamed et al., 2007*; *Zhang et al., 2016*; *Bamidele & Jiang, 2019*).

The analysis of the distribution of the EAEC genes showed that the most frequent gene in our isolates was *aapA*, followed by *aggR*, and *aatA*. Despite this, the *aatA* gene was the only EAEC gene that was more frequent in isolates obtained from symptomatic patients. This gene codes for the protein dispersin, which has a role in the formation of a surface coat that mediates bacterial dispersion by avoiding the aggregation by the aggregative fimbriae (*Sheikh et al., 2002*). It is probable that, in the hybrid UPEC/EAEC isolates, it also contributes to the dispersion and helps in the establishment of a symptomatic infection. This could be accomplished with the presence of UPEC virulence genes, for example, *cfn1* and *hlyA*, which we also found to be more frequent in isolates from symptomatic patients, and the three genes were determined to increase the risk of developing pyelonephritis. In fact, *cnf1* and *hlyA* have been shown to play a role in inflammation, and the deletion of

either one attenuates cystitis mediated by UPEC strain CP9 (*Smith et al., 2015*). Both genes have been shown to be strongly associated in urinary strains (*Marrs, Zhang & Foxman, 2005*), however only 13 of our isolates had this combination. The interplay of *cnf1*, *hlyA* and *aatA* in the development of clinical symptoms and pyelonephritis needs to be addressed in more detail.

Biofilm formation is an important characteristic of EAEC and UPEC strains. It is involved not only in the persistence of the infections, but also contributes to bacterial resistance to antibiotics (*Hicks, Candy & Phillips, 1996*; *Soto, 2014*). In this regard, most of the isolates obtained in this work were low biofilm producers, and only 22.11% were high producers. Accordingly, biofilm production was not found to be related to a specific infection or the risk of development. Neither we found differences in biofilm production between UPEC/EAEC and UPEC isolates.

Regarding antibiotic resistance, UPEC and UPEC/EAEC isolates showed a similar antibiotic resistance profile, except for antibiotics cefazoline, ceftriaxone, cefepime, aztreonam, tobramycin and ciprofloxacin; and a higher number of EAEC isolates were ESBL producers (77.27%). Also, more UPEC/EAEC isolates (90.91%) were MDR than UPEC isolates (63.64%). This percentage is higher than those found in EAEC isolates obtained from patients with intestinal infections in two different studies (75% and 58%) (*Hebbelstrup Jensen et al., 2018*, *Chattaway et al., 2017*). This suggests that these isolates are more likely to acquire antibiotic resistance genes. To our knowledge, this is the first report that analyses the antibiotic resistance in UPEC isolates containing EAEC genes. More studies are needed to identify and monitor these bacterial isolates, which should be relevant in clinical practice due to its potential pathogenic effect.

## CONCLUSIONS

UPEC/EAEC isolates are more frequent in symptomatic patients and the *aatA* gene was associated with a higher risk of developing pyelonephritis, along with UPEC genes *hlyA* and *cfn1*. UPEC/EAEC isolates obtained from UTI showed ESBL production and MDR.

## ACKNOWLEDGEMENTS

We thank José Gatica Bello for statistical analysis.

### Funding

This work was supported by the Fondo FOMIX-CONACYT Gobierno del Estado de Guerrero, Convocatoria M0008 2014-01 (No. 249671). The funders had no role in study design, data collection and analysis, decision to publish, or preparation of the manuscript.

### Grant Disclosures

The following grant information was disclosed by the authors:
The Fondo FOMIX-CONACYT Gobierno del Estado de Guerrero, Convocatoria M0008 2014-01: 249671.

## Competing Interests

The authors declare there are no competing interests.

## Author Contributions

- Verónica I. Martínez-Santos and Arturo Ramirez- Peralta analyzed the data, authored or reviewed drafts of the paper, and approved the final draft.
- María Ruíz-Rosas conceived and designed the experiments, prepared figures and/or tables, and approved the final draft.
- Oscar Zaragoza García analyzed the data, prepared figures and/or tables, and approved the final draft.
- Luis Armando Resendiz-Reyes and Obed Josimar Romero-Pineda performed the experiments, prepared figures and/or tables, and approved the final draft.
- Natividad Castro-Alarcón conceived and designed the experiments, authored or reviewed drafts of the paper, supervised experimental work, and approved the final draft.

## Human Ethics

The following information was supplied relating to ethical approvals (*i.e.*, approving body and any reference numbers):

The study was approved by the Research Ethics Committee of the Autonomous University of Guerrero (CB-002/2021) and the Ethics Committee of the ISSSTE clinic.

## Data Availability

The dataset with the raw results and patients' information is available in the Supplemental File.

## Supplemental Information

Supplemental information for this article can be found online at http://dx.doi.org/10.7717/peerj.11726#supplemental-information.

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
