# Peer review of "Enteroaggregative Escherichia coli is associated with antibiotic resistance and urinary tract infection symptomatology"

_PeerJ, doi:10.7717/peerj.11726_

## Round 0.1 · original submission · Minor Revisions

The manuscript was evaluated by two reviewers and found merit to be published in this Journal. Please address their concerns accordingly.

Reviewer 1 ·

Basic reporting

No comment

Experimental design

no comment

Validity of the findings

no comment

Additional comments

Urinary tract infections (UTIs) are among the most common bacterial infections in humans. Among bacterial species involved in UTIs, uropthogenic Escherichia coli (UPEC) is the most common. Antimicrobial resistance in UPEC and spreading MDR UPEC in recent decades is a clinical problem. Increasing frequency of MDR UPEC result in excessive use of broad-spectrum antibiotics such as fluoroquinolones, cephalosporins and aminoglycosides. Antimicrobial resistance among UPEC is increasing in many countries and shows time- and area-related variability.
This manuscript can be published in Peer J but minor revision is necessary:
1. Line 47: Instead 105 CFU should be ≥105CFU
2. Line 116: Please state which cards dedicated to Vitek2 have been used to identify bacteria, determine antibiotic susceptibility and detect resistance mechanisms.
3. Instead “strains” should be “isolates”. Change “strain” to “isolate”, please correct this throughout the manuscript.
4. Line 174: Change “Phylogenetic grouping …. “ to “Phylogenetic groups and presence of EAEC isolates among UPEC”
5. Line 182: Change “Pathotypes and population characteristics” to “Occurrence of UPEC and UPEC/ EAEC in patients with different clinical symptoms and demographic characteristics”
6. Line 198: Change “Virulence genes …..” to “Occurrence of virulence genes in EAEC and UPEC isolated from asymptomatic and symptomatic patients”.
Line 216: Change “susceptibility pattern” to “resistance”.

Reviewer 2 ·

Basic reporting

this manuscript is clear and unambiguous. data are appropriate as presented. figures and tables are OK. English is good.

Experimental design

The question is well-defined and the is rigorous. However, there are some gaps that should be addressed, e.g. what is the frequency of the reported genes in fecal E. coli isolates?

Validity of the findings

The findings appear valid but the reservations below.

Additional comments

This is a useful contribution to the field. but i have some criticisms the response to which will strengthen teh paper.

1. although EAEC genes appear to be prevalent (surprisingly so), it would make a more powerful case for their association with UTI if their frequencies among fecal strains in teh same human population were known. i doubt they could be as high as among the urinary isolates, but this is a critical point for the ms.

2. AggR controls expression of both aatA and aap genes. therefore, one would expect that there would be a high degree of correspondence among the genes. Indeed, presence of aap or aatA without aggR would suggest that the former loci would be silent, and therefore not clinically relevant (unless another regulator were at play). Moreover, the Aat apparatus was initially reported to be the transporter of the aap gene produce dispersin, so they should also be linked (in theory). the authors should report the combinations of genes - perhaps a supplemental table of the 100 isolates showing all their genotypes would be helpful for the circumspect reader.

3. Table 4. i'm not sure what the p values represent.

4. line 188: "no", rather than "none"

---

## Round 0.2 · accepted · Accept

The authors have properly addressed all the concerns previously raised by the Reviewers.